# Positive Point-of-Care Influenza Test Significantly Decreases the Probability of Antibiotic Treatment during Respiratory Tract Infections in Primary Care

**DOI:** 10.3390/diagnostics13122031

**Published:** 2023-06-12

**Authors:** Aneta Rzepka, Anna Mania

**Affiliations:** 1Medicus Primary Health Care Centre, Magdalena Kurnatowska, ul. Starogostyńska 9, 63-800 Gostyń, Poland; rzepkaaneta@hotmail.com; 2Department of Infectious Diseases and Child Neurology, Poznan University of Medical Sciences, 60-572 Poznań, Poland

**Keywords:** respiratory tract infections, primary care, diagnostic tools, decision making, comorbidities

## Abstract

This study aimed to analyse clinical and laboratory findings in primary care patients with respiratory tract infections to distinguish the group more likely to receive antibiotic treatment. The study group consisted of 631 patients (264 males; 367 females) with a median age of 48 years (IQR 36–63 years). Analysed groups included patients treated with antibiotics (n = 269 patients; 43%) and those who recovered without antibiotic treatment (n = 362 patients; 57%). Patients receiving antibiotics were older (median 51 vs. 47 years; *p* = 0.008) and more commonly developed fever (77% vs. 25%, *p* < 0.0001) and cough (63% vs. 30%; *p* = 0.0014). Moreover, they more frequently presented wheezing and crackles upon physical examination (28% vs. 4% and 9% vs. 0.3%; *p* < 0.0001 and *p* < 0.0001, respectively). They also had more comorbidities and came to more follow-up visits (median of 4 vs. 3 and 2 vs. 1, *p* < 0.0001 and *p* < 0.0001, respectively). Patients receiving symptomatic therapy more often had positive point-of-care tests (POCTS)—20% vs. 7%; *p* = <0.0001. Multivariate analysis in our cohort found comorbidities complexity (odds ratio—OR 2.62; 95% confidence interval—1.54–4.46), fever (OR 32.59; 95%CI 19.15–55.47), crackles (OR 26.35; 95%CI 2.77–250.81) and the number of visits (OR 4.15; 95%CI 2.39–7.20) as factors increasing the probability of antibiotic treatment. Positive influenza POCTS reduced the risk of antibiotic therapy (OR 0.0015; 95%CI 0.0001–0.0168).

## 1. Introduction

Respiratory tract infections (RTIs) comprise the most common causes of medical advice in primary care. They constitute 10–15% of all visits during the year. In temperate climates, RTIs may be responsible for most visits from late autumn to early spring [1]. The majority of patients suffer from upper respiratory tract infections (URTIs). However, lower respiratory tract infections (LRTIs) usually have a more severe clinical course [2]. In most patients, RTIs are of viral aetiology and have a self-limiting character. The most common viruses causing RTIs include rhinoviruses, coronaviruses, respiratory syncytial virus, influenza, parainfluenza, metapneumovirus and adenoviruses [2]. Conditions such as sinusitis, otitis or unspecified acute URTI require symptomatic treatment in most cases [3].

The potential diagnosis is usually based on reported symptoms, including fever, nasal discharge, sore throat, cough, and hoarseness. Some symptoms and signs are not specific and overlap with bacterial infections caused by *Streptococcus pyogenes*, *Streptococcus peumoniae*, *Moraxella catharralis*, *Hemophilus influenzae* and other species [2,3]. Therefore, access to point-of-care testing (POCT) detecting viral antigens with the result available after a short period seems to be a reliable tool in the diagnostic process. POCT constitute rapid, easy-to-use tests performed by non-laboratory-trained personnel. Before the pandemic of the coronavirus disease 2019 (COVID-19), POCTS were less widely available than they currently are. They may comprise multiple tests for numerous viral pathogens or be restricted to selected ones. The results of rapid influenza and streptococcal antigen tests strongly influence RTI treatment, as the physician may implement a correct antiviral or antibiotic where appropriate [4,5,6]. Nevertheless, positive results of other viral pathogen tests may spare the use of unnecessary medications. Therefore, POCT could be a valuable tool to guide the proper management of patients and infection control measures in primary care [4,5,6].

Influenza is associated with high morbidity and mortality, causing 10% of RTIs with frequent hospital admissions [7,8]. The influenza virus invades and damages epithelial cells of the nose, larynx, trachea and bronchi. The disease is a highly contagious RTI characterised by sudden onset. Typical symptoms include fever, chills, muscle pain, headache, cough and nasal discharge [7,8]. The greatest risk of complications in the course of influenza involves children < 5 years and patients > 65 years of age and pregnant, chronically ill and immunocompromised patients. Typical complications of influenza comprise otitis media, pneumonia, myositis, myocarditis and neurological complications with meningitis, acute disseminated encephalitis and Gullain–Barre syndrome. If diagnosed early, optimally up to 2 days after the onset of symptoms, influenza may be treated more effectively [9,10].

Many patients receive antibiotics because of their severe clinical conditions, lack of improvement of clinical state, or severe complications developed despite the supportive treatment [11]. Decisions made in primary care should be based on the patient’s physical appearance, risk factors present in patient history, basic laboratory parameters, and the results of rapid POCT offered on-site to identify respiratory tract pathogens [12]. The benefit of POCT is an additional aid to physicians to manage patients’ expectations for antibiotics and encourage patients to self-care when suffering from a self-limiting condition. Specific factors that influence the decision-making process in primary care include uncertainty about the exact diagnosis or treatment, worries about past patient experience with the treatment of infections, fear of legal problems if the patient deteriorates or fear of being perceived as having achieved nothing for patients. All of these factors could be reduced using POCTS [13].

Antibiotics should be prescribed when needed to treat RTIs in primary care, and the benefits of their use in managing RTIs are marginal in most diagnoses. Inappropriately prescribing antibiotics is common. Available reports suggest that approximately 30–52% of outpatient antibiotic prescriptions are unnecessary [14,15]. An improper antibiotic prescription may increase the prevalence of antibiotic-resistant bacterial strains, one of the greatest threats to public health worldwide [16]. Consequently, standard antibiotic treatments are no longer effective, making severe bacterial infections harder to control.

Moreover, antibiotics may have numerous side effects, including *Clostridium difficile* infection. Over half of the antibiotic expenditure occurs in outpatient settings [17]. Several antibiotic stewardship methods are directed at clinicians to educate, improve the quality and influence public expectations for the antibiotic treatment of RTI [18,19].

Antibiotic stewardship interventions for the judicious use of antimicrobials for RTI have significantly reduced the prescribing of antibiotics in primary care [20,21]. Numerous aspects have not been adequately assessed yet, including the impact on RTI cases in which antibiotic treatment is usually indicated [22].

This study aimed to analyse if using POCT could influence the level of antibiotic prescription and identify other clinical and laboratory findings in primary care patients with respiratory tract infections to distinguish the group more likely to receive antibiotic treatment.

## 2. Materials and Methods

### 2.1. Study Design and Participants

The study involved 631 consecutive patients from primary care consulted in a single centre due to RTIs from January 2019 to March 2020. Participation in the study was voluntary and confirmed with the patient’s written consent. The study was designed as a prospective observational cohort study following the World Medical Association Declaration of Helsinki and reporting guidelines for strengthening the reporting of observational studies in epidemiology (STROBE).

Inclusion criteria included symptoms of RTIs (fever, sore throat, sneezing, hoarseness, cough, nasal discharge, dyspnoea, tachycardia and tachypnoea).

The sample size was calculated using an online calculator (calculator.net) with a confidence level of 95%, a margin of error of 5%, an estimated proportion of patients receiving antibiotics of 35% and a population of 1000 patients in the family practice. The estimated sample size was 260. All patients with the symptoms of RTIs who attended a primary care practice were approached and asked to participate in the study. According to the study protocol, patients who declined to take part would be excluded from the study with their data not included in the analysis. However, all of them agreed to contribute.

### 2.2. Assessments

An observational cohort approach was used for data collection. A detailed history was taken of all patients. Collected clinical data included comorbidities, smoking habits and chronic treatment. Comorbidities complexity was established if five underlying conditions were diagnosed. A thorough physical examination followed the interview and collection of historical data. All findings were put in the clinical records of the patients. Diagnoses were based on clinical presentation and reported symptoms. Clinical scales (i.e., the Centor scale) were used to evaluate the probability of bacterial infection. Laboratory tests were requested by a physician when clinically necessary to establish the final diagnosis or potential bacterial or viral aetiology. They included a complete blood count with differential in 333 patients and inflammatory parameters, namely C-reactive protein (CRP) and the erythrocyte sedimentation rate (ESR), in 261 patients. URTIs included acute pharyngitis, nasopharyngitis, acute sinusitis or laryngitis and uncomplicated influenza. LRTIs comprised acute bronchitis, pneumonia, cases of complicated influenza and the exacerbation of asthma or COPD.

All patients had clinical symptoms suggestive of influenza, i.e., sudden onset fever +≥1 of the following symptoms: Cough, sore throat, rhinitis or a feeling of blocked nose, ≥1 general systemic symptom: Headache, muscle pain, sweating or shaking, tiredness lasting <72 h. If symptoms resembled influenza, a rapid antigen test was performed. In total, 356 patients (56% of the cohort) underwent rapid influenza tests with an actim Influenza A&B test (Medix Biochemica, Finnland) from the nasopharyngeal swab (sensitivity of 77%, specificity of 95%). Patients with the above symptoms and a positive antigen test result were diagnosed with influenza. One patient was diagnosed with influenza based on clinical symptoms and previous contact with a person with confirmed influenza, despite a negative antigen test result.

### 2.3. Treatment Approach

Based on the patients’ conditions, physical findings and the results of laboratory tests, the decision regarding therapy was made by a family medicine specialist. Patients with suspected viral infections were given supportive treatment. Patients with positive flu-test results received oseltamivir if diagnosed early (<2 days from the onset of symptoms or the patient belonged to the severe influenza course risk group). In contrast, patients with suspected bacterial infections received antibiotics. Based on this decision, we divided the patients into two groups: Treated with antibiotics (n = 269) and managed with symptomatic treatment (n = 362). Both groups exceed the number of the estimated sample size.

### 2.4. Statistical Analysis

The statistical analysis was performed with MedCalc ® Statistical Software version 20.011 (MedCalc Software Ltd., Ostend, Belgium; https://www.medcalc.org; 2021) The quantitative variables were analysed initially for normal distribution with the Shapiro–Wilk test. After the rejection of a normal distribution, further analysis was performed using the Mann–Whitney test. The qualitative variables were compared with the chi-squared test. The values of *p* < 0.05 were deemed statistically significant. Further analysis was performed using logistic regression. Parameters with a statistically significant difference were included in the univariate analysis. Parameters significant in the univariate analysis were included in the multivariate analysis. Thus, parameters without significance were excluded from the model until only significant parameters remained. The results were presented as crude (univariate analysis) or adjusted (multivariate analysis) odds ratio (OR) that were appropriate and a 95% confidence interval (95%CI). Results with 95%CI not including 1.0 were considered statistically significant.

The study received consent from the Bioethical Committee dated 10 January 2019 at Poznan University of Medical Sciences, resolution number 84/19.

## 3. Results

### 3.1. Patient Characteristics

The study group consisted of 631 patients (264 males; 367 females) diagnosed with RTIs with a median age of 48 years (IQR 36–63 years). Fever was present in 333 patients (53% of the study cohort). A positive influenza POCT was obtained in 90 patients out of the 356 tests performed (swabbing rate 56%; positivity rate 25%). Sixty-nine patients (77% of those diagnosed with influenza) received oseltamivir. Moreover, 560 (88%) patients suffered from at least one comorbidity. There was a complex medical background (5 or more comorbidities) in 148 (23%) patients. The median number of comorbidities was 2 (IQR 2–5). The baseline group characteristics are presented in Table 1. URTIs were diagnosed in 492 (78%) patients, and 182 patients developed LRTIs (29%).

### 3.2. Analysis of the Factors in the Group Treated with Antibiotics and Symptomatic Therapy

Baseline clinical parameters were compared in patients treated with antibiotics (n = 269 patients; 43% of the study cohort) and those who recovered without antibiotic treatment (n = 362 patients; 57% of the study cohort). Patients receiving antibiotics were slightly older (median of 51 vs. 47 years; *p* = 0.008) and more commonly developed fever (77% vs. 25%, *p* < 0.0001) and cough (63% vs. 30%; *p* = 0.0014). Moreover, they more frequently presented wheezing and crackles upon physical examination than the patients not receiving antibiotics (28% vs. 4% and 9% vs. 0.3%; *p* < 0.0001 and *p* < 0.0001, respectively). They also had more comorbidities and came to more follow-up visits (median of 4 vs. 3 and 2 vs. 1, *p* < 0.0001 and *p* < 0.0001, respectively). A complex medical background was observed more frequently in the antibiotic-receiving group (32% vs. 17%, *p* < 0.0001).

No significant differences regarding analysed laboratory parameters were noticed between the groups.

In the univariate analysis, we confirmed the differences between the groups regarding fever, the number of comorbidities and comorbidities complexity, diagnoses of URTIs and LRTIs, number of visits, cough frequency, wheezing and crackles. This analysis also confirmed the less likely administration of antibiotics in patients with positive flu tests.

Clinical parameters of patients treated with and without antibiotics are presented in Table 2.

### 3.3. Point-of-Care Influenza Testing

A positive flu test was found more often in the group with supportive treatment than in patients receiving antibiotics (20% vs. 7%; *p* < 0.0001). Out of all 356 swabbed patients, antibiotics were introduced in 150 patients (42%), and 18 had positive influenza tests (20%). In the patients with negative influenza tests (n = 266), antibiotics were administered in 132 (50%). The odds ratio (OR) for receiving antibiotic treatment with positive influenza POCT was 0.25 (95% confidence interval CI; 0.14–0.44). The OR for administering an antiviral treatment with positive influenza POCT was 870.71 (95%CI 115.10–6586.00). The proportion of patients receiving antibiotics and antivirals regarding influenza test results is presented in Figure 1.

### 3.4. Comorbidities

Upon analysing the prevalence of comorbidities in both groups, we found that patients with specific diagnoses received antibiotics more frequently. These underlying illnesses included obesity (14% vs. 6%; *p* < 0.0001), diabetes (19% vs. 12%; *p* < 0.0001), hypertension (47% vs. 34%; *p* = 0.001), inflammatory bowel disease (10% vs. 5%; *p* = 0.022), headaches (45% vs. 34%; *p* = 0.005), epilepsy (4% vs. 1%; *p* = 0.027), degenerative spine conditions (41% vs. 28%; *p* = 0.0008), chronic ischemic heart disease (3% vs. 2%; *p* = 0.0016) and chronic obstructive pulmonary disease (13% vs. 5%; *p* = 0.0003). Moreover, patients who smoked were more often prescribed antibiotics (27% vs. 16%; *p* = 0.0014). The difference was also significant in the univariate analysis. However, the diagnoses of specific comorbidities were not proven to be essential factors influencing the risk of antibiotic treatment in multivariate analysis. The prevalence of comorbidities in groups treated with and without antibiotics is shown in Table 3.

In multivariate analysis, comorbidities complexity, fever and crackles and a higher number of visits were proven to be the risk factors of antibiotic treatment, while positive influenza POCTs significantly decreased the risk of implementation of antibiotics. The results of the multivariate analysis are presented in Table 4.

## 4. Discussion

Although most RTIs are of viral origin requiring supportive treatment, they are the reason for most antibiotic prescriptions in primary care [22]. General use of POCT may help to establish the diagnosis of viral disease. In the case of influenza, patients may benefit more from prompt administration of antiviral treatment in patients with a risk of severe course and complications of the disease [4,6]. We have proven the flu test to be the most helpful in avoiding unnecessary antibiotic therapy in our cohort. The positive test result enabled adequate management of influenza with oseltamivir in most patients. The study was conducted in the pre-COVID-19 period; therefore, only flu tests were included in diagnostic procedures due to the low availability of other POCTs in primary care. Many rapid multi-viral respiratory microbiological POCTs have recently been introduced in primary care. Available reports state that their use may enhance the quality of antibiotic prescribing by reversing the decisions based on clinical findings alone [4,6,11,23]. Comparing pre- and post-test diagnoses suggests the overdiagnosis of suspected bacterial infections and under-diagnosis of influenza without POCT results [23]. An accurate diagnosis of RTI is essential. Symptoms such as fever or cough may be present in influenza, other viral disorders and bacterial infections [2].

However, using POCTS in febrile patients may help distinguish those with influenza and other viral diseases, depending on the test used, from those with different aetiology [11,23,24]. In our cohort, 25% of preformed influenza POCTs were positive. POCTs are valuable in preventing the inappropriate administration of antibiotics in primary care and enabling the proper management of RTIs. Although supplementary diagnostic tools may add effort, increase cost and require trained personnel, directed therapy is more effective and prevents long-term consequences of inappropriate RTI management [6,25]. POCTs have limited value due to their lower sensitivity (77% in the case of the test used in our cohort) than testing based on a polymerase chain reaction (PCR). Many patients with negative POCTs still have RTI caused by viral infection. Moreover, antigen tests are designed to detect pathogens in the acute phase of the disease when the viral load is highest.

Nevertheless, thanks to POCTs, a substantial proportion of patients may benefit from the rapid detection of viral infection to establish a correct diagnosis. Therefore, POCTs seem to be an adequate tool in the primary care setting regarding positive results. Patients with clinical symptoms suggestive of a specific viral disease and negative initial POCT may gain from repeated POCT or PCR testing [26]. Of course, identifying viral pathogens does not rule out concurrent infection with bacterial pathogens. Thus, even the presence of respiratory bacteria in nasopharyngeal swabs does not distinguish between the infection and colonisation without an input of clinical symptoms, which would be essential to determine a correct diagnosis [27]. This fact could explain the administration of antibiotics in 18 patients with positive flu tests in the analysed group.

Inappropriate antibiotic administration may lead to the development of antibiotic-resistant bacterial strains, which is a significant concern for public health worldwide. Nevertheless, patients that require proper treatment should be identified promptly and treated according to their clinical state and approved standards [18,19,20,21,22,23]. Regarding RTIs, specific diagnoses (i.e., streptococcal pharyngitis or bacterial pneumonia) should always be followed by antibiotic treatment. Thus, antibiotic therapy should be restricted to necessary cases in numerous clinical states such as sinusitis, tracheitis, otitis, bronchitis or unspecified RTI [2,11]. The decision on antibiotics prescription in primary care is based on historical data, physical findings and basic laboratory tests when clinically appropriate. Nevertheless, it was established that physicians would prescribe antibiotics when faced with uncertainty in patients’ diagnoses, treating immunocompromised or older patients with comorbidities and for patients demanding antibiotics, especially under time constraints [22,23]. In such cases, a physician could use reliable POCT results as a guide for themselves and proof of a proper diagnosis for the patient. Our study analysed medical history factors and clinical findings to distinguish the group more likely to receive antibiotic treatment in primary care.

In our cohort, the overall antibiotic prescription rate was 43%. Antibiotics were prescribed to 50% of patients with negative influenza test results and 20% with positive influenza POCTs. Antiviral treatment with oseltamivir was used in 77% of patients with positive influenza POCT. Positive influenza POCT significantly reduced the risk of antibiotic treatment and increased the probability of antiviral therapy. Available reports describe similar rates stressing that the guidelines may not support up to 40% of antibiotic use in primary care [24,28,29]. In our study, antibiotics were prescribed to older patients with more chronic comorbidities or greater comorbidities complexity, who came more times for follow-up visits. These factors were also proven by other authors [12,21,29]. Patients with respiratory, cardiovascular, neurological, metabolic, genetic and other comorbidities are at higher risk of more severe clinical course and potential complications in many RTIs. Therefore, patients with an increased risk of RTI complications receive antibiotics more frequently than remaining patients [12,21,30,31]. In the current study, patients with obesity, diabetes, hypertension, chronic obstructive pulmonary disease, chronic bowel diseases, degenerative spinal conditions, epilepsy and headaches received antibiotics more frequently, which was also confirmed in univariate analysis. However, none of the specific comorbidities was proven to be a risk factor for more frequent antibiotic treatment in the multivariate analysis performed in our study.

Delayed antibiotic treatment is one of the advised approaches in primary care. It was proven that the risk of infectious complications from common URTIs is low and not modified by antibiotic treatment. Moreover, patients diagnosed with URTI in whom antibiotics were withheld had an increased 30-day risk of severe infections [32]. Nevertheless, a patient who returns for a subsequent visit with no improvement is more likely to obtain an antibiotic prescription eventually, as was also proven in our study [33,34].

Clinical symptoms more frequently observed in the study group receiving antibiotics included fever and cough, while physical findings comprised wheezing and crackles. Perceived severity of the illness and abnormal results in physical examination predict antibiotics prescription. Such prescribing is considered rather appropriate [35,36]. Fever may be present in RTIs of both viral and bacterial aetiology. As stated previously, a patient with RTI symptoms could benefit from POCT to establish a proper diagnosis. Prolonged fever raises concerns in patients and physicians. However, other studies frequently associated these symptoms with antibiotic administration [12]. Cough is a very nonspecific symptom in viral and bacterial infections.

Nevertheless, a combination of abnormalities indicating LRTI may lead to more frequent administration of antibiotics. In our cohort, diagnosis of LRTI was found to be more frequent in patients treated with antibiotics. This factor was also proven significant in univariate analysis. However, multivariate analysis did not support this finding. Eventually, conclusively differentiating between bacterial and viral causes of RTIs based on signs and symptoms only is rarely possible. Physician concern regarding missed bacterial diagnosis leads to administering antibiotics, which could also be reduced using POCT [37,38].

Blood tests offer precision for diagnosing and managing many diseases and are available in primary care. The results improve decision-making in individual patients [37,38,39]. In our study, however, basic laboratory tests such as cellular blood count with a differential and CRP value did not differ significantly between the groups. Available reports show that point-of-care procalcitonin reduces the probability of antibiotic prescription in approximately 26% of low-risk patients with LRTI in primary care and significantly lowers the risk of antibiotic-related side effects [40,41]. The use of CRP in primary practice was associated with similar results, and overall benefits balance potential harms [37,38,39]. However, the use of CTP-POCT did not influence the clinical recovery, resolution of symptoms and hospital admissions when compared to the usual care [42,43,44].

We conducted our study in a single primary care centre before the COVID-19 pandemic. However, the study period included two subsequent seasons of respiratory tract infections, which allowed for observing a substantial number of patients with RTIs. Our study, although single centre, was performed prospectively. Therefore, we managed to collect complete data in the study cohort. The patients with RTIs were not selected; all patients agreed to participate and signed informed consent. The diagnoses based on physical findings are challenging to verify. Moreover, they may reflect patient management. The fact that the physicians knew about the study could influence the outcome. Nevertheless, the observational study reflects the real-life situation in daily practice in primary care. Therefore, we consider the results valuable.

Physicians often prescribe antibiotics inappropriately when faced with time pressures and patient demands despite their preference for evidence-based practice. Clinical decision support tools, guidelines and patient education programs could help reduce unnecessary antibiotic use [37,38,45,46]. POCT may enormously facilitate the proper diagnosis and management of viral disorders. They are also useful tools to avoid adverse events and unnecessary costs [47].

## 5. Conclusions

In conclusion, POCT for influenza may improve patient management regarding early diagnosis, enabling proper antiviral treatment and saving inappropriate antibiotic therapy. Nevertheless, patients in severe clinical conditions with abnormalities in physical examination suggesting potential bacterial infection should be treated according to suspected aetiology.

## Figures and Tables

**Figure 1 diagnostics-13-02031-f001:**
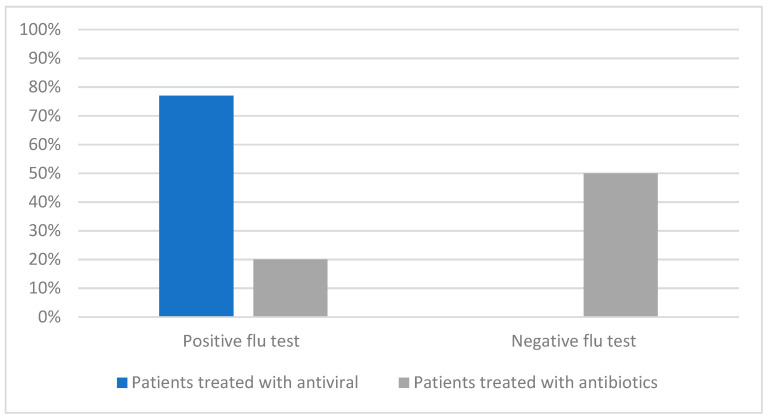
The proportion of swabbed patients treated with antiviral and antibiotics subsequent influenza point-of-care testing (POCT).

**Table 1 diagnostics-13-02031-t001:** Baseline group characteristics.

Feature	Number of Patients	Median	IQR	95%CI
Age (years)	631	48	36–63	46–51
Sex M/F	264/367			
Fever Y/N	333/298			
Flu-test P/N	90/266			
Number of comorbidities	631	3	2–5	3–3
Comorbidities complexity Y/N	148/484			
URTI Y/N	492/139			
LRTI Y/N	182/449			
Number of visits	632	2	1–2	2–2
WBC (G/L)	333	7.1	5.4–8.8	5.89–7.80
Neutrophils (G/L)	333	3.8	2.85–5.81	3.28–4.30
Lymphocytes (G/L)	333	2.3	1.90–2.85	2.17–2.60
CRP (mg/dL)	261	19	11–30	15.0–21.81
ESR (mm/h)	261	20	12–28	15–22

M—male, F—female, Y—Yes, N—No, URTI—upper respiratory tract infection, LRTI—lower respiratory tract infection, WBC—white blood count, CRP—C-reactive protein, ESR—erythrocyte sedimentation rate, IQR—interquartile range, 95%CI—confidence interval.

**Table 2 diagnostics-13-02031-t002:** Comparison of clinical and laboratory parameters in patients receiving and not receiving antibiotics.

Feature	Patients Receiving Antibiotics			Patients Not Receiving Antibiotics			*p*	OR (95%CI)
	Number of Patients	Median	IQR	Number of Patients	Median	IQR		
Age (years)	269	51	38–66	362	47	35–62	0.008 *	1.01 (1.01–1.04)
Sex M/F	111/158			153/209			0.778	-
Fever Y/N	208/61			90/272			<0.0001 *	10.31 (7.11–14.94)
Flu test P/N	18/132			72/134			<0.0001 *	0.26 (0.14–0.44)
Number of comorbidities	269	4	2–6	362	3	1–4	<0.0001 *	2.19 (1.51–3.19)
Comorbidities complexity Y/N	85/184			63/299			<0.0001 *	2.19 (1.51–3.19)
Number of visits	269	2	1–3	362	1	1–2	<0.0001 *	2.15 (1.74–2.66)
URTI Y/N	193/76			299/63			0.001	0.53 (0.36–0.78)
LRTI Y/N	98/171			84/278			0.0003 *	1.89 (1.34–2.68)
Cough Y/N	169/100			111/151			0.0014 *	1.69 (1.23–2.33)
Throat pain Y/N	154/115			201/161			0.67	-
Wheezing Y/N	75/194			13/349			<0.0001 *	10.37 (5.61–19.18)
Crackles Y/N	23/246			1/361			<0.0001 *	33.75 (4.52–251.58)
WBC (G/L)	112	8.60	5.55–9.10	221	7.0	5.05–7.75	0.22	-
Neutrophils (G/L)	112	5.85	3.8–7.35	221	3.70	2.62–4.52	0.27	-
Lymphocytes (G/L)	112	2.35	2.01–2.60	221	2.30	1.80–2.95	0.97	-
CRP (mg/dL)	112	19	15.75–30.50	221	10	11.1–26.2	0.55	-
ESR (mm/h)	112	20	11.75–30.00	221	17.5	10.75–20.0	0.48	-

M—male, F—female, Y—Yes, N—No, URTI—upper respiratory tract infection, LRTI—lower respiratory tract infection, WBC—white blood count, CRP—C-reactive protein, ESR—erythrocyte sedimentation rate, IQR—interquartile range, OR = odds ratio, 95%CI—confidence interval; * *p* < 0.05.

**Table 3 diagnostics-13-02031-t003:** Prevalence of comorbidities in the group receiving and not receiving antibiotics.

Feature	Patients Receiving Antibiotics	Patients not Receiving Antibiotics	*p*	OR (95%CI)
	Number of Patients N = 269	Number of Patients N = 362		
Diabetes Y/N	51/218	22/340	<0.0001 *	3.62 (2.13–6.13)
Obesity Y/N	37/232	44/318	<0.0001 *	3.63 (2.13–6.16)
Inflammatory bowel disease Y/N	27/242	19/343	0.022 *	2.01 (1.09–3.70)
Hypertension Y/N	126/143	123/239	0.001 *	1.71 (1.24–2.37)
Cigarette smoking Y/N	72/197	59/303	0.0014 *	3.45 (1.07–11.14)
Hypercholesterolemia Y/N	83/186	88/274	0.07	-
Malignancy Y/N	17/252	15/347	0.22	-
Degenerative spine conditions Y/N	110/159	102/260	0.0008 *	1.76 (1.26–2.46)
Epilepsy Y/N	10/259	4/358	0.027 *	2.15 (1.74–2.66)
Headaches Y/N	121/148	123/239	0.005 *	1.58 (1.15–2.19)
Chronic ischemic heart disease Y/N	20/249	8/354	0.0016 *	3.55 (1.54–8.19)
Chronic obstructive pulmonary disease Y/N	34/235	17/345	0.0003 *	1.89 (1.00–3.57)
Asthma Y/N	22/247	22/340	0.31	-

Y—Yes, N—No, OR—odds ratio, 95%CI—confidence interval; * *p* < 0.05.

**Table 4 diagnostics-13-02031-t004:** Multivariate logistic regression for the probability of receiving antibiotics.

Feature	aOR	95%CI	*p*
Comorbidities complexity	2.62	1.54–4.46	0.0004
Fever	32.59	19.15–55.47	<0.0001 *
Positive flu-test	0.0015	0.0001–0.0168	<0.0001 *
Number of visits	4.15	2.39–7.20	<0.0001 *
Crackles	26.35	2.77–250.81	0.044 *

aOR—adjusted odds ratio, CI—confidence interval; * *p* < 0.05.

## Data Availability

The data presented in this study are available upon request from the corresponding author.

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
