# Peer review of "Positive Point-of-Care Influenza Test Significantly Decreases the Probability of Antibiotic Treatment during Respiratory Tract Infections in Primary Care"

_diagnostics, 2023, doi:10.3390/diagnostics13122031_

Round 1

Reviewer 1 Report

Dear Authors

The paper is very interesting because it deals with a very important health problem.

I would like to make some comments and raise some doubts.

Due to my academic background, I will only focus on the methodology and data analysis part.

Statistical analyses are correct to achieve the objective.

Major Issue

Although I believe that the sample is large enough to have good results, I would like to know if a sample size study was done and how it was done so that this sample is representative of the study population. If this sample size study was not done, I would like to know if a statistical power study was done for a sample of this size.

Minor issue

In the text the term "gender" is used when the variable is "sex". The term "gender" refers to social role and how society views you. For this reason I consider it necessary to use the term "sex" throughout the paper.

In the text they speak that the W Shapiro-Wilk. The result of this test is not mentioned anywhere in the text.

In Table 1, 95%CIs are of the mean or median?

Odd ratios are adjusted, i.e. they are multivariate analysis. They are not specified in the tables.

Author Response

Dear Reviewer,

Thank you very much for all your comments and suggestions. 

The sample size was calculated using an online calculator (calculator.net) with a confidence level of 95%, a margin of error of 5%, an estimated proportion of patients receiving antibiotics of 35% and a population of 1000 patients in the family practice. The estimated sample size was 260. Both groups exceed the number of the estimated sample size. The description was added in the text and marked red.

The quantitative variables were analysed initially for normal distribution with the Shapiro-Wilk test. After rejecting normal distribution, further analysis was performed using the Mann–Whitney test. Since the Shapiro-Wilk test was performed to choose the proper test for further analysis the results were not presented as we consider it would make them unclear. We may add these results in supplementary material if necessary. 

The term gender was corrected to sex according to suggestions.

In Table 1, 95%CIs are of the median as presented.

Odd ratios are crude for univariate analysis and adjusted for multivariate analysis - the issue was cleared in the description of the statistical analysis and in the Tables. 

Reviewer 2 Report

Line 36: word virus is missing in RSV

Line 37: references behind this list of viruses

Line 42: references

Lines 52-61: references

Table 1 & 2: Any major differences between males and females?

Does age and sex play any part?

Overall the study is very light on supporting literature and could benefit from deeper insights from the literature. As it were expand the insights for the benefit of the readers.

English language is good.

Author Response

Dear Reviewer,

Thank you very much for all your comments and suggestions. 

Suggested references were added to the lines in the introduction. They are marked red in brackets and the reference list.

Table 1 presents descriptive statistics of the whole study cohort. Therefore no differences are observed there. In Table 2 slightly significant difference regarding age is marked and described in the text (lines 176-177). No differences regarding sex were observed.

The discussion was expanded regarding the literature review. References were added and marked red.

English was carefully revised.

Round 2

Reviewer 1 Report

Dear authors, thank you very much for answering all my questions and accepting my suggestions to improve the paper.